# Perioperative Hyperspectral Imaging to Assess Mastectomy Skin Flap and DIEP Flap Perfusion in Immediate Autologous Breast Reconstruction: A Pilot Study

**DOI:** 10.3390/diagnostics12010184

**Published:** 2022-01-13

**Authors:** Tim Pruimboom, Anouk A. M. A. Lindelauf, Eric Felli, John H. Sawor, An E. K. Deliaert, René R. W. J. van der Hulst, Mahdi Al-Taher, Michele Diana, Rutger M. Schols

**Affiliations:** 1Department of Plastic, Reconstructive and Hand Surgery, Maastricht University Medical Center, 6200 MD Maastricht, The Netherlands; r.vander.hulst@mumc.nl; 2Department of Cardiothoracic Surgery, Maastricht University Medical Center, 6229 HX Maastricht, The Netherlands; anouk.lindelauf@mumc.nl; 3Department of Visceral Surgery and Medicine, University of Bern, 3010 Bern, Switzerland; eric.felli@dbmr.unibe.ch; 4Department of Biomedical Research, Hepatology, University of Bern, 3012 Bern, Switzerland; 5Department of Plastic, Reconstructive, and Hand Surgery, VieCuri Medical Center, 5912 BL Venlo, The Netherlands; jsawor@viecuri.nl (J.H.S.); adeliaert@viecuri.nl (A.E.K.D.); 6Department of Surgery, Maastricht University Medical Center, 6229 HX Maastricht, The Netherlands; mahdi.altaher@mumc.nl; 7Research Institute Against Digestive Cancer (IRCAD), 67000 Strasbourg, France; michele.diana@ircad.fr; 8Department of General, Digestive and Endocrine Surgery, University Hospital of Strasbourg, 67200 Strasbourg, France; 9ICube Lab, Photonics for Health, University of Strasbourg, 67400 Strasbourg, France

**Keywords:** hyperspectral imaging, reconstructive surgery, breast reconstruction, mastectomy skin, flap, free flap, tissue necrosis, monitoring, clinical study

## Abstract

Mastectomy skin flap necrosis (MSFN) and partial DIEP (deep inferior epigastric artery perforator) flap loss represent two frequently reported complications in immediate autologous breast reconstruction. These complications could be prevented when areas of insufficient tissue perfusion are detected intraoperatively. Hyperspectral imaging (HSI) is a relatively novel, non-invasive imaging technique, which could be used to objectively assess tissue perfusion through analysis of tissue oxygenation patterns (StO_2_%), near-infrared (NIR%), tissue hemoglobin (THI%), and tissue water (TWI%) perfusion indices. This prospective clinical pilot study aimed to evaluate the efficacy of HSI for tissue perfusion assessment and to identify a cut-off value for flap necrosis. Ten patients with a mean age of 55.4 years underwent immediate unilateral autologous breast reconstruction. Prior, during and up to 72 h after surgery, a total of 19 hyperspectral images per patient were acquired. MSFN was observed in 3 out of 10 patients. No DIEP flap necrosis was observed. In all MSFN cases, an increased THI% and decreased StO_2_%, NIR%, and TWI% were observed when compared to the vital group. StO_2_% was found to be the most sensitive parameter to detect MSFN with a statistically significant lower mean StO_2_% (51% in the vital group versus 32% in the necrosis group, *p* < 0.0001) and a cut-off value of 36.29% for flap necrosis. HSI has the potential to accurately assess mastectomy skin flap perfusion and discriminate between vital and necrotic skin flap during the early postoperative period prior to clinical observation. Although the results should be confirmed in future studies, including DIEP flap necrosis specifically, these findings suggest that HSI can aid clinicians in postoperative mastectomy skin flap and DIEP flap monitoring.

## 1. Introduction

Breast cancer is the most common type of cancer in women worldwide [1,2]. Approximately 40% of female patients will undergo a mastectomy [3]. After a skin-sparing mastectomy, reconstructive surgery using either a prosthesis or autologous tissue can significantly improve patients’ quality of life. The majority of patients who opt for autologous breast reconstruction undergo a so-called deep inferior epigastric artery perforator (DIEP) flap reconstruction [4]. Success rates for DIEP flap viability have dramatically improved over the years [5]. However, partial DIEP flap failure, resulting in fat necrosis, remains a problem in approximately 14% of the procedures [6].

Mastectomy skin flap necrosis (MSFN) is another dreaded postoperative complication, which is reported in up to 30% of all cases of immediate breast reconstructions [7]. In addition to the negative effect on the patient’s physical and mental well-being related to such complications, the health care costs for unexpected reoperations and associated inpatient hospitalization charges were reported to be $16,703 per patient with MSFN in the United States [8].

There are several important patient-related risk factors for developing MSFN, including age, body mass index (BMI) over 30 kg/m^2^, smoking, and adjuvant cancer therapy [9]. Additionally, intraoperative factors, including prolonged operating time, blood loss, and immediate breast reconstruction following skin-sparing mastectomy (SSM) and/or nipple-sparing mastectomy (NSM), increase the risk for postoperative MSFN [7,9,10]. Preferably, areas of insufficient flap viability are detected during surgery, allowing for a direct intervention in case of impaired local tissue perfusion in order to prevent reoperation for necrotic tissue debridement [7]. Although tissue perfusion impairment does not always occur during surgery, the time between the onset of ischemia and re-exploration is directly related to the salvage rate of the DIEP flap. If surgical intervention is not performed within twelve hours of onset of ischemia, tissue damage to the flap is expected to be irreversible [11]. Moreover, postoperative mastectomy skin flap necrosis can also lead to a number of challenges, including prolonged wound management, delayed initiation of adjuvant therapy, patient distress, and risk of infection [12]. In order to prevent long-term problems in the postoperative period following hospital discharge, it is important to detect mastectomy skin flap and DIEP flap necrosis as early as possible during the in-hospital postoperative period.

The gold standard for the assessment of flap viability remains the clinical observation of skin color, capillary refill, and temperature combined with blood flow measurements using Doppler ultrasonography. However, such assessments are rather subjective [13,14]. Although a more objective measurement of tissue viability is advocated, no currently available device meets all the criteria for the ideal monitoring technique as described by Creech and Miller in 1975 [15,16].

Hyperspectral imaging (HSI) has gained popularity in the field of surgery over the last decade. HSI is a relatively novel, non-invasive imaging technique, which could be used to monitor tissue oxygenation in a rapid, easy, and objective manner. The first use of HSI to intraoperatively determine tissue perfusion was reported in 2001 [17]. Although tissue perfusion assessment is currently the main indication to perform HSI [18,19,20,21,22,23], it is also used for wound analysis and for the differentiation between healthy skin and skin cancer [24,25]. A hyperspectral camera system provides the possibility of measuring hemoglobin levels, with oxyhemoglobin (O_2_Hb) and de-oxyhemoglobin (HHb) as its derivatives, and to analyze tissue oxygenation patterns (StO_2_%) and near-infrared (NIR%), tissue hemoglobin (THI%), and tissue water (TWI%) perfusion indices [22]. As a result, a tissue oxygenation map can be generated, as it could be of assistance to predict those areas of skin at risk for necrosis.

The aims of this prospective clinical pilot study are to evaluate the efficacy of HSI for the perfusion assessment of mastectomy skin flap and DIEP flap and to identify cut-off values for tissue necrosis in patients undergoing immediate autologous breast reconstruction.

## 2. Materials and Methods

This prospective study was conducted in 10 women who underwent unilateral skin-sparing mastectomy and immediate autologous breast reconstruction using a DIEP flap at VieCuri Medical Center (Venlo, The Netherlands) between February and July 2021. Patients were excluded in case of secondary autologous reconstruction. All procedures in this study were performed in accordance with the ethical standards of the regional research committee (Medical Ethics Committee in Maastricht: METC-2021-2607). Written informed consent was obtained from all patients prior to their enrolment in the study. Preoperative patient characteristics were obtained from the medical records of the included patients. Variables obtained included age, body mass index (BMI), smoking status, the patient’s past medical history (e.g., diabetes mellitus, preoperative chemotherapy, and/or radiotherapy) as well as the patient’s baseline blood pressure and heart rate intraoperatively. Additionally, the intraoperative use of vasopressors was registered. Postoperative flap necrosis (i.e., mastectomy skin flap or DIEP flap) was assessed within 72 h following surgery and followed up during outpatient clinic appointments.

### 2.1. Surgical Procedure (DIEP Flap Reconstruction) and Postoperative Course

The oncologic surgeon, specialized in breast surgery, started the procedure with a skin-sparing (non-nipple-sparing) mastectomy using electrosurgical monopolar cautery to remove breast tissue. In the meantime, the plastic surgeon started to dissect the most suitable deep inferior epigastric artery perforator in the abdomen. Pre-operatively, all patients underwent a magnetic resonance angiography (MRA) of the abdomen to check and select the perforator(s). Additionally, the perforators were marked on the abdomen using a handheld Doppler device, which was also used intraoperatively in order to guide the dissection of the perforator artery and vein.

Once the mastectomy was performed, a rib-sparing technique was applied to dissect the internal mammary artery (IMA) and vein (IMV) in the second or third intercostal space. Following the dissection of the donor and recipient vessels, the vein was anastomosed using a microvascular anastomotic coupler ranging from 2.0 to 3.0 mm in size (Synovis Micro Companies Alliance, Birmingham, AL, USA). The arterial anastomosis was performed in an end-to-end fashion using an Ethilon 9.0 suture (Ethicon, Inc., Somerville, NJ, USA). After shaping and positioning the DIEP flap underneath the mastectomy skin, the majority of the DIEP flap skin was de-epithelialized except for the region where the nipple was previously positioned. This part (i.e., the “skin island”) allowed for postoperative flap monitoring.

After closure of the recipient and donor sites, patients were admitted to the medium-care unit for one night of intensive post-operative observation prior to transfer to the surgical ward for approximately 4 to 5 days before hospital discharge. The DIEP flap was monitored by checking ultrasound Doppler signals, color, temperature, and capillary refill every hour for the first 24 h, every 2 h on the second day, every 4 h on the third day, and every 8 h on the fourth day. Before the operation, patients received intravenous antibiotics, Cefazolin 2000 mg. This dose was repeated after 4 h during surgery. No prophylactic antibiotics were administered postoperatively. However, patients received prophylactic low-molecular-weight heparin until discharge and wore compression socks during the operation up to the time of adequate mobilisation (i.e., walking more than 30 min a day).

### 2.2. Hyperspectral Imaging (HSI) Protocol

In this study, the hyperspectral camera system (TIVITA™, Diaspective Vision GmbH, Am Salzhaff, Germany) was used. The HSI camera and a computer system, including the monitor, were mounted on a mobile medical cart to allow bedside evaluation in the operation room and on the ward. The imaging technique was previously described circumstantially [22]. In short, HSI is based on the assessment of contiguous spectra (i.e., light of different wavelengths) individually re-emitted via molecules, whereby the molecule-specific re-emitted wave spectrum is generated based on the light spectrum of the halogen spotlights initially emitted via the camera. These data are then processed by computerized algorithms, specific for the molecule of interest, specifically hemoglobin, oxygenated hemoglobin, and water. During HSI, 5 images are recorded and analyzed in a time period of approximately 6 s, including an RGB (red, green, and blue) color image, and an additional 4 images. The 4 images represent the following parameters: tissue oxygenation with a penetration depth of 1 mm (StO_2_%: 0–100%), near-infrared perfusion index with a penetration depth of 4 to 6 mm (NIR% as arbitrary units: 0–100), and the distribution of hemoglobin as tissue hemoglobin index (THI%, arbitrary units: 0–100) and tissue water index (TWI%, arbitrary units: 0–100).

The HSI camera is fixed to a stable and mobile arm to ensure a constant distance of 50 cm above the surgical field (i.e., the breast), represented by 2 separate indicator light points (red and green), which turn orange when in an overlapping position. The measuring area was approximately 20 by 30 cm; the standard image size was 640 by 480 pixels.

During this study, a total of 19 hyperspectral images per patient were recorded. The first HSI assessment of the breast skin was performed preoperatively in the outpatient clinic. The second HSI acquisition was performed following mastectomy, the third one following 10 min during which the internal mammary artery (IMA) was temporarily clamped, the fourth one after completion of the DIEP flap reconstruction, and the fifth to the sixteenth HSI every hour over a postoperative duration of 12 h. HSI number 17, 18, and 19 were performed 24, 48, and 72 h postoperatively, respectively. During the second and third HSI, the mastectomy skin flap was positioned over the chest wall in a tension-free fashion, with 2 to 3 pieces of moist gauze underneath the flap to mimic the previously removed breast volume.

To allow for quantification of perfusion parameters (i.e., StO_2_%, NIR%, THI%, and TWI%) in the investigated area, the camera-specific software package (TIVITA™ Suite) allows to define circular regions of interest (ROI) in which the mean value for each parameter is calculated [22,26].

Three ROIs were manually positioned in the areas of interest, namely the vertical scar underneath the DIEP flap skin island to the inframammary fold (1), the skin island of the DIEP flap, where the nipple was previously positioned (2), and the medial side of the mastectomy skin flap (3) (Figure 1a). Clinical evaluation of the flap was performed to assess the vitality of the tissue (i.e., vital versus necrotic DIEP flap or mastectomy skin flap).

### 2.3. Statistical Analysis

Continuous variables were reported as mean ± standard error (SEM). Statistical analysis was performed with Prism GraphPad 9. A *p*-value < 0.05 was considered statistically significant.

Student *t*-test, Mann–Whitney U test and One-Way analysis of variance (ANOVA) were used following the analysis of data distribution. Principal Component Analysis (PCA) was performed with the following continuous variables StO_2_%, THI%, TWI%, and time (in hours). Data were labeled depending on the diagnosis with categorical classification (“vital” and “necrosis”). The principal component analysis (PCA) was based on the standardization of data and on the largest eigenvalues. The cut-off value for StO_2_% was calculated using a simple logistic regression classifying data into necrosis (0) or vital (1).

## 3. Results

Ten women with a mean age of 55.4 years underwent immediate autologous breast reconstruction using a unilateral DIEP flap following SSM. Table 1 summarizes the main characteristics of the patients included in the current study. Two patients had undergone breast radiotherapy following previous breast-conserving surgery years before the current skin-sparing mastectomy.

In this study, DIEP flap (ROI-2) necrosis and necrosis of the medial side of the mastectomy skin flap (ROI-3) was not observed. However, three patients presented with mastectomy skin flap necrosis in ROI-1. In these patients, the first clinical signs of necrosis were observed at 1 day (i.e., 24 h), 6 days, and 13 days following the operation. Two of the three patients recovered with conservative wound treatment, and one patient was re-operated for debridement and primary closure of the wound. None of the three patients had a history of breast radiotherapy. However, one of the three patients had undergone chemotherapy prior to mastectomy. The mean DIEP flap weight in the group of necrotic mastectomy skin flaps (i.e., necrosis group) of 819 ± 244.7 g was higher when compared to the vital group (726.3 ± 103.9 g). However, this difference was not statistically significant (*p*-value = 0.302). Regarding HSI perfusion parameters, the first hour following the operation, the mean tissue saturation index (StO_2_%) in all ROIs was approximately 45 to 55% in the vital group. In the necrosis group, a mean StO_2_% of 31.93% (SEM 1.04) was observed in the postoperative period prior to clinical observation of skin flap necrosis. Concerning the near-infrared index (NIR%) and the tissue water index (TWI%), the mean values in the necrosis group (approximately 35–45% and 40–45%, respectively) were also lower when compared to the vital group (approximately 25% and 25%, respectively). On the other hand, the mean tissue hemoglobin index (THI%) was found to be lower in the vital group (15 to 45%) as compared to the necrosis group (45 to 80%) (Figure 1).

The mean preoperative (PO), post-mastectomy (PM), and post-clip (PC) perfusion parameters in ROI-1, ROI-2, and ROI-3 in the vital group were compared to the parameters in the necrosis group in order to evaluate tissue perfusion before the flap inset and to evaluate the potential influence of the consecutive steps of the surgical procedure. No statistically significant difference was found. Accordingly, there were no differences in baseline measurements between the groups (Figure 2).

The effect of temporarily clamping the IMA on skin perfusion of the medial side of the mastectomy skin was evaluated by comparing HSI perfusion parameters before and after clamping the IMA (Appendix A). No statistically significant difference in perfusion was found.

As previously described, no necrosis was observed in ROI-2 and ROI-3. Overall (i.e., in both groups), all four tissue parameters were comparable over time. When comparing the necrosis group to the vital group in ROI-2 and ROI-3, only minor differences were observed. Interestingly, the tissue hemoglobin index (THI%) was higher for the necrosis group when compared to the vital group in ROI-2 and lower in ROI-3, respectively.

Hyperspectral images of all parameters of both groups are displayed in Figure 3a. Postoperatively, mastectomy skin flap necrosis is represented by an increased THI%, a decreased oxygen perfusion (StO_2_% and NIR%), and a decreased TWI%. As shown in Figure 3b, oxygen perfusion decreases immediately in the necrosis group, with a statistically significant lower value (−34.81%) 2 h postoperatively when compared to the vital group (*p*-value = 0.0329). The other parameters were found to be less sensitive when compared to StO_2_% and showed no statistically significant difference between the two groups. PCA performed with the HSI parameters shows the cases of necrosis clustered in the positive side of the PC1 (Figure 3c). The hypothesis-driven analysis of data distribution of StO_2_% only for both groups are displayed in Figure 3d. The vital group showed a higher overall StO_2_% when compared to the necrosis group (51 versus 32%, respectively, *p* < 0.0001). A cut-off value of 36.29% was found (95%CI 33.79–38.60%).

## 4. Discussions

This prospective clinical pilot study demonstrated the added value of perioperative HSI in the assessment and early detection of flap necrosis in immediate autologous breast reconstruction. A standard clinical assessment of flap (e.g., mastectomy skin flap, free flap) viability following breast reconstructive surgery is rather subjective with considerable inter-observer variability, and it is predominantly performed by the nurse or the residents [13,14]. Based on the findings in this study, we believe that the implementation of HSI incorporates a strong potential to improve the outcomes of reconstructive flap surgery.

The results of this study suggest that HSI is superior in identifying postoperative mastectomy skin flap necrosis when compared to the conventional clinical evaluation in immediate unilateral, skin-sparing breast reconstruction using DIEP flap. HSI allowed to discriminate patients who developed MSFN from patients without flap necrosis. Accordingly, StO_2_% values were significantly lower in the group of patients who developed MSFN (51%) as compared to the vital group (32%). Regarding StO_2_%, a cut-off value for MSFN of 36.29% was found. Below this value, there is a higher chance (i.e., >50%) of tissue necrosis than the tissue remaining vital. In addition, the StO_2_% was predictable for MSFN, as a mean StO_2_% of 31.93% was observed in the postoperative period prior to the first clinical observation of skin flap necrosis in three patients (at days 1, 6, and 13, respectively). As no necrosis of the DIEP flap was observed in the current study, we were unable to define a StO_2_% cut-off value for DIEP flap necrosis specifically. However, we hypothesize that there are no major differences in the cut-off value for DIEP flap necrosis and mastectomy skin flap necrosis, as the technique for tissue assessment is similar in both flaps (i.e., in both cases it concerns the assessment of “skin flaps”). This similarity in tissue assessment is confirmed by the observation of comparable mean HSI values in ROI-2 and ROI-3, representing a vital part of the mastectomy skin flap as well as a vital DIEP flap. Although we believe that HSI could also be useful in DIEP flap monitoring, this hypothesis should be confirmed in future studies including a larger sample size in order to obtain objective data regarding DIEP flap necrosis.

As baseline HSI values were comparable in all ROIs, and since only postoperative HSI values in the patient who developed MSFN in ROI-1 were lower, the reliability of the data was considered high.

As previously described, HSI allows to assess tissue viability using four different parameters, namely tissue oxygenation (StO_2_%), the near-infrared perfusion index (NIR%), the tissue hemoglobin index (THI%), and the tissue water index (TWI%). A study by Kohler et al. reported that StO_2_% and NIR% were more suitable for skin flap perfusion monitoring when compared to THI% and TWI%, as StO_2_% and NIR% provided more information regarding the microcirculation in the superficial layers of the skin [20]. The results of the current study are in congruence with the ones of Kohler et al. since StO_2_% is the most suitable parameter for the assessment of skin viability, with an accuracy of 93%. NIR% values also decreased mastectomy skin flap necrosis cases. However, the slope of the decrease was not as steep as the one of StO_2_. This could be explained by the penetration depth of both parameters. NIR% is measured in a tissue depth of 4 to 6 mm, whereas StO_2_% is measured at a tissue depth of 1 mm. Since this is more superficial, it provides more information on the perfusion of the mastectomy skin flap, which is overlying the DIEP flap [20].

The current study aimed to identify cut-off values for tissue necrosis and found a cut-off value for StO_2_% of 36.29%. Although there are very few articles in the literature regarding the relation between the HSI parameter value and tissue necrosis, the results of the current study are in agreement with the cut-off value of 40% for StO_2_% and NIR% values reported in two earlier studies regarding postoperative skin flap monitoring with the use of HSI [20,21]. In one patient in the current study, low saturation values (mean StO_2_% of 42% and mean NIR% of 32%) were observed in the region of clinically observed skin ecchymosis. However, no necrosis was observed during follow-up. For this reason, it would be preferable to use a range of parameter values as reported by Jafari-Saraf et al. [27]. Accordingly, a StO_2_% value of 50% or higher represents a “normal wound healing,” a value of 30 to 50% represents a grey area which indicates the need for an extra clinical evaluation, and a value below 30% represents a “bad wound healing” due to a poor perfusion [27]. The aim of outlining the StO_2_% values is to avoid unnecessary reoperations in the “grey area,” ranging from 30 to 50%.

Another method to assess flap necrosis is near-infrared fluorescence angiography using indocyanine green (ICG) [12]. A major advantage of HSI over fluorescence angiography is that HSI does not require the intravenous administration of a contrast agent. Regarding the use of different types of tissue oximeters in which flap viability is measured with sensors placed on the patient, previous studies reported different cut-off values for sufficient tissue saturation [28,29]. For instance, Repez et al. reported a cut-off value of 50% for regional tissue oxygen saturation (rStO_2_%) when using the InSpectra Model 352 (Hutchinson Technology Inc., Hutchinson, MN, USA) with a reported penetration depth of 12 to 13 mm [29]. Keller et al. reported a lower cut-off value of 30% for tissue oxygenation when using the ViOptix (ViOptix Inc., Fremont, CA, USA) [28]. In addition, a study by Lindelauf et al. reported a cut-off value of approximately 50% when using the foresight (Edwards Lifesciences, Irvine, CA, USA) to measure the rStO_2_% [30].

The penetration depth of the ViOptix (1 to 10 mm) is almost comparable to the TIVITA™ device (1 to 6 mm). Additionally, a cut-off value of 30% is currently accepted as a cut-off value for StO_2_% values in reconstructive surgery [31,32]. However, there are several clinical oximeters commercially available with different working mechanisms for each device. For instance, different devices use different wavelengths and sensors, a different ratio between arterial and venous saturation to calculate the rStO_2_%, as well as a different penetration depth [33]. Consequently, Hyttel-Sorensen et al. studied the absolute rStO_2_% values of three different devices and concluded that it is not possible to compare the absolute values for different monitoring systems [34]. We therefore agree with the previous study of our group that it would be better to define device-specific cut-off values than to generalize values [30].

It is essential to combine all four parameters to determine whether the decrease in StO_2_% and NIR% values is caused by an arterial inflow or a venous outflow problem. In case StO_2_ and NIR% both decrease and THI% increases, the problem is related to the venous outflow. In case of a decrease in all the three parameters, the problem is most probably related to the arterial supply. In addition, in case of an increase in THI% and StO_2_% and unchanged NIR% and TWI% values, one should think of a superficial hematoma [21]. In the current study, all cases of necrosis were caused by a venous outflow problem (low StO_2_% and NIR% and high THI%).

Despite the novelty of this prospective clinical pilot study reporting the applicability and efficacy of hyperspectral imaging in assessing mastectomy skin flap and DIEP flap viability, some limitations need to be addressed. First of all, this study was limited by a small sample size. However, 19 TIVITA™ images per patient were taken in a time window of 72 h postoperatively, which resulted in a large dataset that made it possible to closely monitor postoperative changes. We are aware of the need for studies with a larger sample size to confirm the efficacy of HSI in assessing mastectomy skin flap and DIEP flap viability and determine the exact timing of tissue debridement based on HSI instead of clinical evaluation.

Additionally, no continuous HSI measurements were collected to assess tissue perfusion during follow-up. However, we believe that the frequency applied in this study is clinically relevant and logistically feasible. Previous studies also concluded that hyperspectral imaging systems could well detect a decrease in oxygenation before clinical evaluation and Doppler assessment [20,21,22]. For this reason, continuous measurement is unnecessary. In addition, postoperative measurements were performed during conventional postoperative moments of clinical evaluation. This method is in line with the current clinical management of free flaps and will support the assessment without creating extra work in continuously monitoring values.

When considering the availability and costs of the TIVITA™ system, the device is available for purchase in every continent with an estimated total cost of $40,000 [21]. Following purchase, the costs per HSI assessment are negligible since only electricity is required. Up to date, no cost-effectiveness studies have been published.

In conclusion, we demonstrated that HSI could have the potential to accurately assess mastectomy skin flap and DIEP flap viability and discriminate between patients with and without necrosis during the early postoperative period. Although the results of this pilot study should be interpreted with caution given the small sample size, we believe that the results provide useful information to improve the outcomes of reconstructive flap surgery. Although the results should be confirmed in future studies including DIEP flap necrosis specifically, these finding support the hypothesis that HSI can aid clinicians in postoperative flap monitoring, as it, based on the results of this study, enables the detection of necrosis earlier than clinical evaluation. HSI has the potential to facilitate early and accurate debridement of necrotic tissue during the in-hospital postoperative period in order to prevent long-term wound problems in the postoperative period. Ultimately, clinicians should be able to debride necrotic tissue based on HSI assessment, preferably during initial surgery and at least during the same hospital admission before clinical observation of necrosis.

## Figures and Tables

**Figure 1 diagnostics-12-00184-f001:**
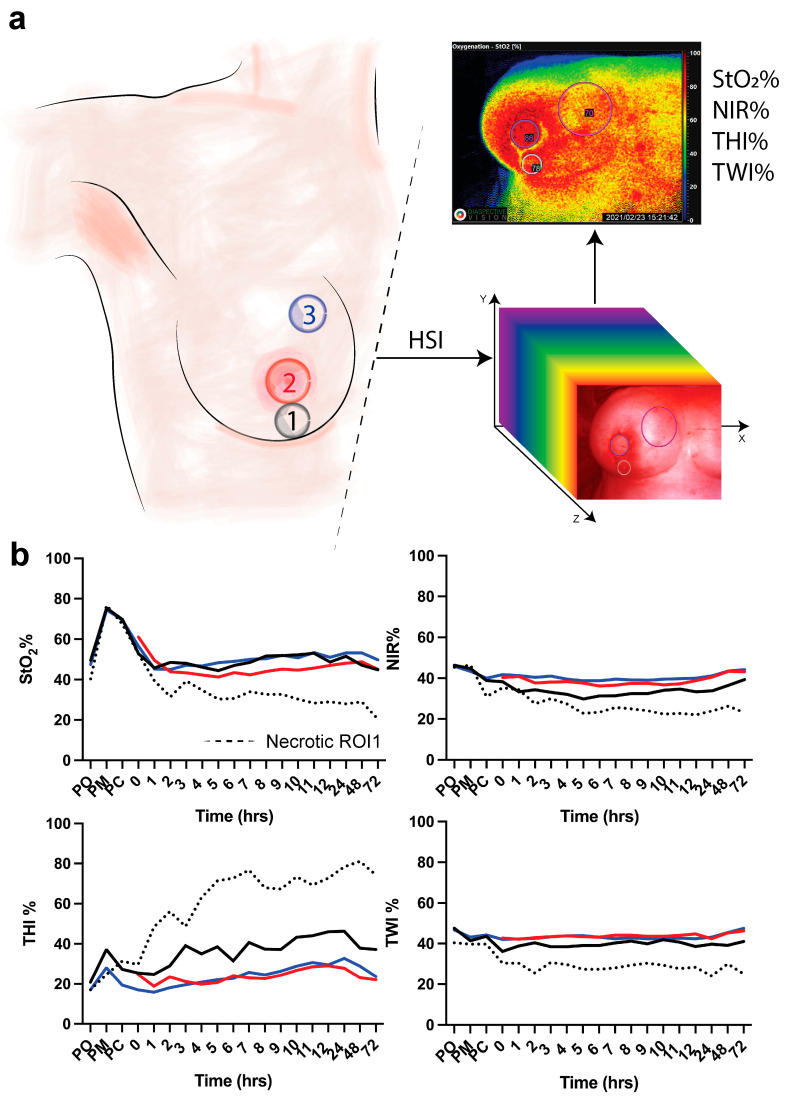
(**a**) Regions of interest. ROI-1 (black) is the region of the vertical scar underneath the skin island of the DIEP flap to the inframammary fold; ROI-2 (red) is the skin island of the DIEP flap. ROI-3 (blue) is the medial side of the mastectomy skin flap. Hyperspectral imaging (HSI) is performed to extract the hypercube with which the algorithm extracts 4 parameters (StO_2_%, NIR%, THI%, and TWI%). (**b**) HSI acquisition is made pre-operatively (PO), post-mastectomy (PM), post-clip (PC), and every hour for 12 h and every day for 3 days. Data are expressed as mean value.

**Figure 2 diagnostics-12-00184-f002:**
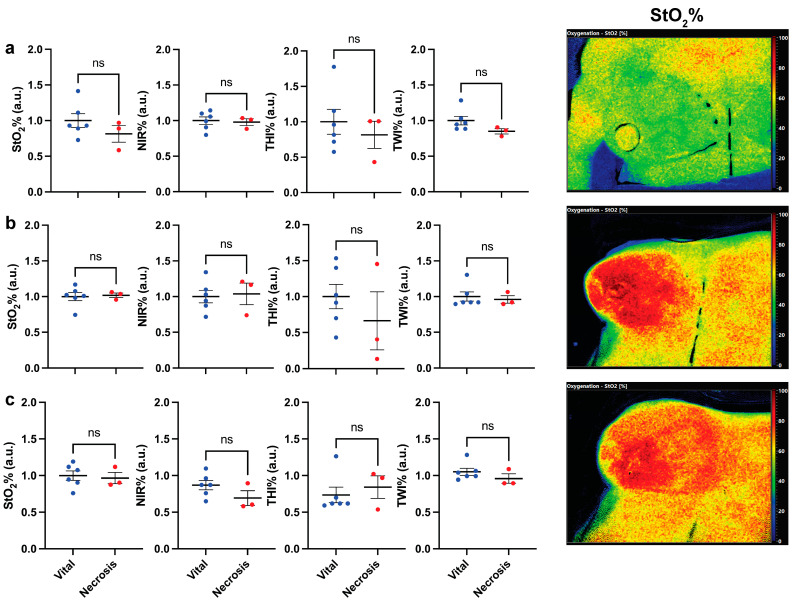
Analysis of perfusion (HSI) parameters between the vital and necrotic mastectomy skin flaps (**a**) pre-operatively (PO), (**b**) post-mastectomy (PM), and (**c**) post-clip (PC). On the left, cumulative statistics of all patients; on the right side, hyperspectral images of a patient at the corresponding timepoint. Data are expressed as mean ± SEM. Student *t*-test was used; *p* > 0.05 was considered non-significant (ns).

**Figure 3 diagnostics-12-00184-f003:**
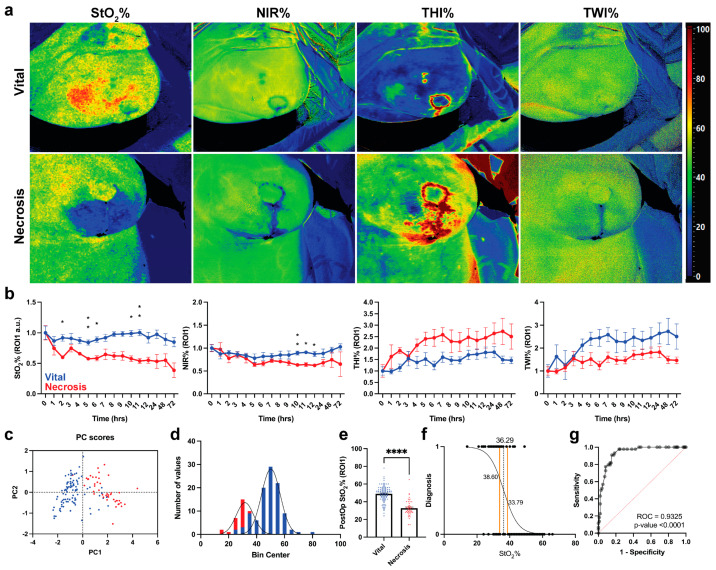
HSI data analysis for necrotic and vital mastectomy skin flaps. (**a**) Visualization of every parameter in both groups. (**b**) Statistical analysis of every parameter in ROI-1. Data are expressed as mean ± SEM and analyzed with One-Way ANOVA mixed effect. *p*-Value < 0.05 was considered significant. Data are normalized with the respective parameter values immediately after the operation and expressed as fold change. (**c**) PCA shows that necrotic mastectomy skin flaps are clustered on PC1 positive values. (**d**) Data distribution of necrotic and vital mastectomy skin flaps. (**e**) Mann–Whitney U test of StO_2_% values of necrotic mastectomy skin flaps (*n* = 47 measurements) versus vital skin flaps (*n* = 104 measurements) of ROI-1. Data are expressed as mean ± SEM. A two-tailed *p*-value < 0.05 was considered significant. ****: *p* < 0.0001. (**f**) Simple logistic regression of StO_2_% values according to the following classification: necrotic: 0, vital: 1. (**g**) ROC of simple logistic regression shown in (**f**).

**Table 1 diagnostics-12-00184-t001:** Patient and surgical characteristics. Data are expressed as mean ± standard error (SEM). BMI, body mass index.

	Mean ± SEM	*n*
Patients		10
Age (years)	55.4 ± 1.6	
Weight (kg)	81.8 ± 4.5	
Length (cm)	168 ± 1.9	
BMI (kg/m^2^)	29 ± 1.8	
DIEP flap weight (grams)	757.2 ± 98.6	
Operative time (minutes)	428 ± 10.9	
Cup breast size		
B		2
C		1
D		4
E		1
F		2
Smoking status		
Yes		0
No		10
Diabetes mellitus		
Yes		0
No		10
ASA classification		
1		1
2		3
3		0
4		0
Previous chemotherapy		
Yes		1
No		9

## Data Availability

The data underlying the results presented in this paper are not publicly available at this time but may be obtained from the authors upon reasonable request.

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
