# Peer review of "Perioperative Hyperspectral Imaging to Assess Mastectomy Skin Flap and DIEP Flap Perfusion in Immediate Autologous Breast Reconstruction: A Pilot Study"

_diagnostics, 2022, doi:10.3390/diagnostics12010184_

Round 1
Reviewer 1 Report
this study has a well structured research design. However, the sample size is quite small, and in my opinion it is not adequate to draw any conclusion.
The authors should highlight the clinical relevance of their conclusions.
“In addition, HSI values were predictable for MSFN as low HSI values were observed in a mean period of seven days prior to the first……”
Which HIS values are referred here??
“In conclusion, in this study, we demonstrated that HSI could accurately assess flap viability” The study was focused on mastectomy flaps viability. This should be clarified throughout the entire manuscript.
Moreover, the title should be modified accordingly.
Which is the relevance of detecting necrosis of mastectomy skin flaps prior to clinical observation should be highlighted.
Conclusion section
“In conclusion, in this study, we demonstrated that HSI could accurately assess flap viability and discriminate between patients with and without necrosis during the early postoperative period. This provides useful information to improve the outcomes of reconstructive flap surgery, in terms of facilitating early and accurate debridement of necrotic tissue to prevent potential long-term wound problems in the postoperative period.”
Do the authors suggest to perform debridement before necrosis is clinical evident? This should be clarified
Author Response
Answers to reviewer 1
Thank you for your comments. All your comments have been separately addressed, please see our reply in the attached Word file (changes in the text are underlined). The changes are highlighted as ‘tracked changes’ in the manuscript.
Thank you for your comments. All your comments have been separately addressed, please see our reply below (changes in the text are underlined). The changes are highlighted as ‘tracked changes’ in the manuscript.
- This study has a well structured research design. However, the sample size is quite small, and in my opinion it is not adequate to draw any conclusion.
As we already mentioned in the discussion section, we agree that the sample size is small. To our opinion, 19 hyperspectral images per patient over time together with the pre-operative measurements, resulted in a dataset of an adequate size in which we could define no differences in baseline measurements and distinguish necrotic from vital tissue. However, we think that the findings of this study, including the reported cut-off value for necrosis should be interpreted with caution.
For this reason, we changed the following sentences in the conclusion section: “In conclusion, in this study, we demonstrated that HSI could have the potential to accurately assess mastectomy skin flap and DIEP flap viability and discriminate between patients with and without necrosis during the early postoperative period. Although the results of this pilot study should be interpreted with caution given the small sample size, we believe that the results provide useful information to improve the outcomes of reconstructive flap surgery.” See Discussion: page 11, paragraph 3.
Furthermore, we changed the following sentence in the discussion section to: “The results of this study suggest that HSI is superior in identifying postoperative mastectomy skin flap necrosis when compared to the conventional clinical evaluation in immediate unilateral, skin-sparing breast reconstruction using DIEP flap.” See Discussion: page 9, paragraph 2.
Moreover, the text in the abstract section was changed to: “It was concluded that HSI could has the potential to accurately assess tissue perfusion and discriminate between vital and necrotic skin flap during the early postoperative period prior to clinical observation of skin flap necrosis.” See abstract: page 1.
- The authors should highlight the clinical relevance of their conclusions.
To further emphasize the clinical relevance, we have changed the following sentence in the conclusion section to: “These findings support the hypothesis that HSI can aid clinicians in postoperative flap monitoring, as it, based on the results of this study, enables the detection of necrosis earlier than clinical evaluation. HSI has the potential to facilitate early and accurate debridement of necrotic tissue during initial hospital admission in order to prevent long-term wound problems in the postoperative period.” See Discussion: page 11, paragraph 3.
Moreover, we added the following sentence to the abstract section: “These findings suggest that HSI can aid clinicians in postoperative flap monitoring.” See Abstract: page 1.
- “In addition, HSI values were predictable for MSFN as low HSI values were observed in a mean period of seven days prior to the first……” Which HSI values are referred here??
We understand that this sentence is confusing as information on the qualification (i.e., StO2%, NIR%, TWI% or THI%) and quantification (percentage of perfusion parameter, 0-100%) of the HSI value is missing. In this sentence we refer to the StO2% of the three patients who developed mastectomy skin flap necrosis, at day 1, day 6, and day 13 (mean of 7 days) following the operation, diagnosed with clinical evaluation. Accordingly, we changed this sentence to: “In addition, the StO2% was predictable for MSFN as a mean StO2% of 31.93% was observed in the postoperative period prior to the first clinical observation of skin flap necrosis in three patients (at day 1, 6 and 13, respectively).” See page 9, paragraph 2.
In addition, we changed the following sentence concerning StO2% in the results section to: “In the “necrosis group”, a mean StO2% of 31.93% (SEM 1.04) was observed in the postoperative period prior to clinical observation of skin flap necrosis.” See Results: page 6, paragraph 1.
- “In conclusion, in this study, we demonstrated that HSI could accurately assess flap viability” The study was focused on mastectomy flaps viability. This should be clarified throughout the entire manuscript.
The study focused on both DIEP flap viability and mastectomy skin flap viability. Since no DIEP flap necrosis, and only mastectomy skin flap necrosis was observed during this study, more emphasis has been placed on the mastectomy skin flap. However, both flaps (mastectomy skin flap and DIEP flap) were assessed. For clarification, we changed the following sentence in the conclusion section: “In conclusion, we demonstrated that HSI could have the potential to accurately assess mastectomy skin flap and DIEP flap viability and discriminate between patients with and without necrosis during the early postoperative period.” See Discussion: page 11, paragraph 3.
- Moreover, the title should be modified accordingly.
Since both the mastectomy skin flaps and DIEP flaps have been assessed using HSI, with flap necrosis only observed in the mastectomy skin flap, we changed the manuscript title to: “The ability of perioperative hyperspectral imaging to detect assess mastectomy skin flap and DIEP flap perfusion necrosis in immediate autologous breast reconstruction”.
- Which is the relevance of detecting necrosis of mastectomy skin flaps prior to clinical observation should be highlighted.
When concerning the DIEP flap, it is important to detect early signs of necrosis as early as possible, since the time between the onset of ischemia and re-exploration is directly related to the successful salvage rate of the flap. If the surgical intervention is not performed within twelve hours of onset of ischema, tissue damage to the flap is expected to be irreversible [1]. In addition, when mastectomy skin flap necrosis is diagnosed early following surgery (preferably during initial hospital admission), long-term problems, including prolonged wound management, delayed initiation of adjuvant therapy, patient distress and risk of infection, can be prevented. Therefore, the following information including the reference [11] is added to the introduction section: “Although tissue perfusion impairment does not always occur during surgery, the time between the onset of ischemia and re-exploration is directly related to the salvage rate of the DIEP flap. If surgical intervention is not performed within twelve hours of onset of ischemia, tissue damage to the flap is expected to be irreversible [11]. Moreover, postoperative mastectomy skin flap necrosis can also lead to a number of challenges, including prolonged wound management, delayed initiation of adjuvant therapy, patient distress and risk of infection [12]. In order to prevent long-term problems in the postoperative period following hospital discharge, it is important to detect mastectomy skin flap and DIEP flap necrosis as early as possible during the initial hospital admission.” See Introduction page 2, paragraph 3.
- Irwin MS, Thorniley MS, Doré CJ, Green CJ (1995) Near infra-red spectroscopy: a non-invasive monitor of perfusion and oxygenation within the microcirculation of limbs and flaps. BJPS 48(1):14-22. https://doi.org/10.1016/0007-1226(95)90024-1
- “In conclusion, in this study, we demonstrated that HSI could accurately assess flap viability and discriminate between patients with and without necrosis during the early postoperative period. This provides useful information to improve the outcomes of reconstructive flap surgery, in terms of facilitating early and accurate debridement of necrotic tissue to prevent potential long-term wound problems in the postoperative period.” Do the authors suggest to perform debridement before necrosis is clinical evident? This should be clarified
Based on the findings of our study, it seems that HSI could facilitate early and accurate debridement of necrotic tissue during initial hospital admission, prior to clinical observation of flap necrosis. However, given the small sample size of this pilot study, we are aware of the need for studies with a larger sample size to confirm the efficacy of HSI in assessing mastectomy skin flap and DIEP flap viability and determine the exact timing of tissue debridement based on HSI instead of clinical evaluation. This information is added to the Discussion section: “We are aware of the need for studies with a larger sample size to confirm the efficacy of HSI in assessing mastectomy skin flap and DIEP flap viability and determine the exact timing of tissue debridement based on HSI instead of clinical evaluation”. See Discussion: page 10, paragraph 5.
In addition, we added the following sentence to the conclusion section: “Ultimately, clinicians should be able to debride necrotic tissue based on HSI assessment, preferably during initial surgery and at least during the same hospital admission before clinical observation of necrosis.” See Conclusion: page 11, paragraph 3.
Reviewer 2 Report
The authors are to be commended on perioperative hyperspectral imaging to detect flap necrosis using 4 parameters combined technology to assess the flap vascularity.
I think that this study is possible to have an important role in detecting flap necrosis using perioperative imaging. However, this study’s design is not enough to support the authors’ conclusion.
Some questions to be answered are below:
- As the authors mentioned in discussion section (Line 343), this study’s limitation is its’ very small sample size. Even as a pilot study, this study’s sample of 10 patients (19 images/patient) are not enough to distinguish partial DIEP necrosis, because the authors referred that partial DIEP flap necrosis is occurred in only 14 %. and there is no partial DIEP flap necrosis in this sample. In contrast, this study includes 3 mastectomy skin flap necrosis in 10 patients, because mastectomy skin flap necrosis is frequently occurred as mentioned (line52). How the authors try to evaluate not only mastectomy skin flap necrosis but also partial DIEP necrosis risk using 10 patients’ data. The authors need to add some patients in the study group to mention partial DIEP flap necrosis or need to eliminate the assessment on partial DIEP necrosis risk.
- The authors noted that the golden standard for the assessment of flap viability remain the clinical observation. Why does the study have no control group to evaluated the usefulness of the hyperspectral imaging.
- Conclusion is too strong from the 10 patients’ case series, because the authors do not show apparent improvement of the outcome in reconstructive surgery by using their modality.
Author Response
Answers to reviewer 2
Thank you for your comments. All your comments have been separately addressed, changes in the text are underlined. The changes are highlighted as ‘tracked changes’ in the manuscript.
The authors are to be commended on perioperative hyperspectral imaging to detect flap necrosis using 4 parameters combined technology to assess the flap vascularity. I think that this study is possible to have an important role in detecting flap necrosis using perioperative imaging. However, this study’s design is not enough to support the authors’ conclusion.
Thank you for your comments. All your comments have been separately addressed, please see our reply below (changes in the text are underlined). The changes are highlighted as ‘tracked changes’ in the manuscript.
- As the authors mentioned in discussion section (Line 343), this study’s limitation is its’ very small sample size. Even as a pilot study, this study’s sample of 10 patients (19 images/patient) are not enough to distinguish partial DIEP necrosis, because the authors referred that partial DIEP flap necrosis is occurred in only 14%. and there is no partial DIEP flap necrosis in this sample. In contrast, this study includes 3 mastectomy skin flap necrosis in 10 patients, because mastectomy skin flap necrosis is frequently occurred as mentioned (line52). How the authors try to evaluate not only mastectomy skin flap necrosis but also partial DIEP necrosis risk using 10 patients’ data. The authors need to add some patients in the study group to mention partial DIEP flap necrosis or need to eliminate the assessment on partial DIEP necrosis risk.
We agree that it is not possible to evaluate (partial) DIEP necrosis in a small sample of 10 patient with no single observation of DIEP flap necrosis. On the other hand, as all of the DIEP flaps survived with no observation of (partial) DIEP flap necrosis, we can evaluate the observed HSI values as sufficient perfusion (i.e., vital DIEP flap). As this was a pilot study to test the ability of HSI to assess perfusion assessment of both the mastectomy skin flap and DIEP flap and, fortunately for the patients, no DIEP flap necrosis was observed, we changed the manuscript title to: “The ability of perioperative hyperspectral imaging to detect assess mastectomy skin flap and DIEP flap perfusion necrosis in immediate autologous breast reconstruction”.
Since the technique for tissue assessment is similar for perfusion assessment of the mastectomy skin flap and DIEP flap and the mean HSI values in ROI-2 and ROI-3 were comparable, representing a vital part of the mastectomy skin flap as well as a vital DIEP flap, we expect no major differences in the cut-off value for DIEP flap necrosis. Therefore, we added the following sentence to the discussion section: “Although we were unable to define a StO2% cut-off value for DIEP flap necrosis specifically, we expect no major differences in the cut-off value for DIEP flap necrosis and mastectomy skin flap necrosis as the technique for tissue assessment is similar in both flaps (i.e. in both cases it concerns the assessment of “skin flaps”). This hypothesis is strengthened by the observation of comparable mean HSI values in ROI-2 and ROI-3, representing a vital part of the mastectomy skin flap as well as a vital DIEP flap.” See Discussion; page 9, paragraph 2.
- The authors noted that the golden standard for the assessment of flap viability remain the clinical observation. Why does the study have no control group to evaluated the usefulness of the hyperspectral imaging.
We agree that the golden standard for the assessment of flap viability is clinical evaluation of the mastectomy skin flap and/or DIEP flap. Every patient served as its own control in this study as the HSI values were compared to clinical evaluation, therefore we believe that an extra control group only evaluation tissue with clinical evaluation is redundant.
- Conclusion is too strong from the 10 patients’ case series, because the authors do not show apparent improvement of the outcome in reconstructive surgery by using their modality.
We agree that the conclusion is too strong in a pilot study including 10 patients. For this reason, we changed the following sentences in in the conclusion section: “In conclusion, in this study, we demonstrated that HSI could have the potential to accurately assess mastectomy skin flap and DIEP flap viability and discriminate between patients with and without necrosis during the early postoperative period. Although the results of this pilot study should be interpreted with caution given the small sample size, we believe that the results provide useful information to improve the outcomes of reconstructive flap surgery.” See Discussion: page 11, paragraph 3.
Furthermore, we changed the following sentence in the discussion section to: “The results of this study suggest that HSI is superior in identifying postoperative mastectomy skin flap necrosis when compared to the conventional clinical evaluation in immediate unilateral, skin-sparing breast reconstruction using DIEP flap.” See Discussion; page 9, paragraph 2.
Moreover, the text in the abstract section was changed to: “It was concluded that HSI could has the potential to accurately assess tissue perfusion and discriminate between vital and necrotic skin flap during the early postoperative period prior to clinical observation of skin flap necrosis.” See Abstract; page 1.
Reviewer 3 Report
Thank you very much for the opportunity to review this manuscript. In this study, the authos evaluate through a prospective clinical pilot sthe efficacy of hyperspectral imaging for the perfusion assessment of mastectomy skin flap and Diep Inferior Epigastric Perforator flap, and to identify cut-off values for tissue necrosis in patients undergoing immediate autologous breast reconstruction. This reviewer believes the following comments need to be addressed prior to reevaluating this manuscript:
1- Did you correlate or adjust for flap weight in your analysis?
2- What was the mean operative time?
3- Please comment on the cost of using this technology.
4- Please comment on the availability of this technology.
5- How will using this technology impact your practice?
Thanks again for the opportunity to review this manuscript.
Author Response
Answers to reviewer 3
Thank you for your comments. All your comments have been separately addressed, changes in the text are underlined. The changes are highlighted as ‘tracked changes’ in the manuscript.
Thank you very much for the opportunity to review this manuscript. In this study, the authors evaluate through a prospective clinical pilot the efficacy of hyperspectral imaging for the perfusion assessment of mastectomy skin flap and Diep Inferior Epigastric Perforator flap, and to identify cut-off values for tissue necrosis in patients undergoing immediate autologous breast reconstruction.
Thank you for your comments. All your comments have been separately addressed, please see our reply below (changes in the text are underlined). The changes are highlighted as ‘tracked changes’ in the manuscript.
- Did you correlate or adjust for flap weight in your analysis?
As the sample size of the pilot study is small (N=10) we did not correlate or adjust for flap weight in our analyses. We agree that this is interesting information and, therefore, added the DIEP flap weight to the manuscript. See table 2: Patient and surgical characteristics. Furthermore, we added the following information on the mean flap weight of “vital group” and the “necrotic group” to the results section: “The mean DIEP flap weight in the group of necrotic mastectomy skin flaps (i.e., “necrosis group”) of 819± 244.7 grams was higher when compared to the “vital group” (726.3± 103.9 grams). However, this difference was not statistically significant (p value = 0.302).” See Results: page 5, paragraph 2.
- What was the mean operative time?
The mean operative time was: 428 minutes (SEM 10.9) This information is added to table 2: Patient and surgical characteristics.
- Please comment on the cost of using this technology.
The total costs for purchasing the TIVITA™ hyperspectral imaging system are estimated to be $40.000 [1]. Up do date, no cost effectiveness studies have been published. The following sentence is added to the discussion section: “When considering the availability and costs of the TIVITA™ system, the device is available for purchase in every continent with an estimated total cost of $40.000 [21]. Following purchase, the costs per HSI assessment are negligible since only electricity is required. Up to date, no cost effectiveness studies have been published.” See Discussion: page 11, paragraph 2.
- Schulz, T.; Marotz, J.; Stukenberg, A.; Reumuth, G.; Houschyar, K.S.; Siemers, F. [Hyperspectral imaging for postoperative flap monitoring of pedicled flaps]. Handchir Mikrochir Plast Chir 2020, 52, 316-324, doi:10.1055/a-1167-3089.
- Please comment on the availability of this technology.
This technology is commercially available in every continent. Most studies are performed in Europe or the USA. As previously mentioned, this information is added to the discussion section.
- How will using this technology impact your practice?
This technology has the potential to aid clinicians in assessing tissue perfusion of the mastectomy skin flap and DIEP flap intraoperatively and postoperatively in order to minimize postoperative complications. To further emphasize the clinical relevance, we have changed the following sentence in the conclusion section to: “These findings support the hypothesis that HSI can aid clinicians in postoperative flap monitoring, as it, based on the results of this study, enables the detection of necrosis earlier than clinical evaluation. HSI has the potential to facilitate early and accurate debridement of necrotic tissue during initial hospital admission in order to prevent potential long-term wound problems in the postoperative period.” See Discussion: page 11, paragraph 3.
Furthermore, we added the following sentence to the abstract section: “These findings suggest that HSI can aid clinicians in postoperative flap monitoring.” See Abstract: page 1.
Moreover, we added the following sentence to the conclusion section: “Ultimately, clinicians should be able to debride necrotic tissue based on HSI assessment, preferably during initial surgery and at least during the same hospital admission before clinical observation of necrosis.” See Conclusion: page 11, paragraph 3.
Round 2
Reviewer 1 Report
As the authors stated in Materials and methods this is a pilot study. This should be clearly stated in the title.
the authors described an interesting study on HSI for mastectmoy skin flap and DIEP flap monitoring.
Howevere, Diep flap necrosis was not reported, and no objective data were provided in this study to sustain the usefulness/superiority of HSI in DIEP flap monitoring (as reported in the paragraph below). In my opinion a future resubmission should be focused on skin flap mastectomy monitoring
page 9 "Although we were unable to define a StO2% cut-off value for DIEP flap necrosis specifically, we expect no major differences in the cut-off value for DIEP flap necrosis and mastectomy skin flap necrosis as the technique for tissue assessment is similar in both flaps (i.e., in both cases it concerns the assessment of “skin flaps”). This hypothesis is strengthened by the observation of comparable mean HSI values in ROI-2 and ROI-3, representing a vital part of the mastectomy skin flap as well as a vital DIEP flap."
Diep flap necrosis was not reported, and no objective data were provided by this study to sustain the usefulness/superiority of HSI in DIEP flap monitoring. The entire study should be focused on mastectomy skin flap monitoring.
“it is important to detect mastectomy skin flap and DIEP flap necrosis as early as possible during the initial hospital admission.” Usually hospital admission precedes surgical treatment. Please clarify this statement
“Acceptor site” should be replaced with “recipient site”
“…..and wore pressure stockings during the operation and for the time of (insufficient) immobilization.” Please clarify the meaning of this statement
Author Response
Answers to reviewer 1
Thanks again you for your comments. All your comments have been separately addressed, changes in the text are underlined. The changes are highlighted as ‘tracked changes’ in the manuscript.
As the authors stated in Materials and methods this is a pilot study. This should be clearly stated in the title.
We agree that it should be clear for the readers that this is a pilot study. Therefore, we changed the title to: “Perioperative hyperspectral imaging to assess mastectomy skin flap and DIEP flap perfusion in immediate autologous breast reconstruction: a pilot study”.
The authors described an interesting study on HSI for mastectomy skin flap and DIEP flap monitoring. However, DIEP flap necrosis was not reported, and no objective data were provided in this study to sustain the usefulness/superiority of HSI in DIEP flap monitoring (as reported in the discussion page 9, paragraph 2: “Although we were…vital DIEP flap”). In my opinion a future resubmission should be focused on skin flap mastectomy monitoring. DIEP flap necrosis was not reported, and no objective data were provided by this study to sustain the usefulness/superiority of HSI in DIEP flap monitoring. The entire study should be focused on mastectomy skin flap monitoring.
We agree that no objective data on DIEP flap necrosis is presented, as no DIEP flap necrosis was observed. However, we disagree that the entire study should be focused on mastectomy skin flap monitoring. As previously mentioned in our first ‘answers to the reviewers’, the study focused on both the assessment of DIEP flap perfusion as well as mastectomy skin flap perfusion. Although no DIEP flap necrosis was observed in the current study, we provided objective HSI-data concerning ‘good DIEP flap perfusion’ that did not differ from ‘good mastectomy skin flap perfusion’ in non-necrotic areas. While we still believe in the usefulness of HSI in DIEP flap monitoring, we agree that it is only hypothesis that there is no major difference in the cut-off value for DIEP flap necrosis when compared to mastectomy skin flap necrosis. For this reason, we have the intention to report a future study with larger number of patients in order to add objective data on DIEP flap necrosis. Therefore, we added the following sentence to the discussion: “Although we believe that HSI could also be useful in DIEP flap monitoring, this hypothesis should be confirmed in future studies including a larger sample size in order to obtain objective data regarding DIEP flap necrosis.”. See Discussion: page 9, paragraph 2.
Moreover, we changed the following sentence in the discussion: “As no necrosis of the DIEP flap was observed in the current study, we were unable to define a StO2% cut-off value for DIEP flap necrosis specifically. However, we hypothesize that there are no major differences in the cut-off value for DIEP flap necrosis and mastectomy skin flap necrosis as the technique for tissue assessment is similar in both flaps (i.e., in both cases it concerns the assessment of “skin flaps”). This similarity in tissue assessment is confirmed by the observation of comparable mean HSI values in ROI-2 and ROI-3, representing a vital part of the mastectomy skin flap as well as a vital DIEP flap.” See Discussion: page 9, paragraph 2.
In addition, we changed the following sentence in the discussion to: “Although the results should be confirmed in future studies including DIEP flap necrosis specifically, these finding support the hypothesis that HSI can aid clinicians in postoperative flap monitoring, as it, based on the results of this study, enables the detection of necrosis earlier than clinical evaluation.” See Discussion: page 11, paragraph 3.
The following sentence was also added to the abstract section: “Although the results should be confirmed in future studies including DIEP flap necrosis specifically, these findings suggest that HSI can aid clinicians in postoperative mastectomy skin flap and DIEP flap monitoring.” See Abstract: page 1.
When concerning the title, we deliberately removed ‘the ability of’ from the title and changed it to: “The ability of Perioperative hyperspectral imaging to assess mastectomy skin flap and DIEP flap perfusion in immediate autologous breast reconstruction: a pilot study”
“It is important to detect mastectomy skin flap and DIEP flap necrosis as early as possible during the initial hospital admission.” Usually, hospital admission precedes surgical treatment. Please clarify this statement
We agree that every surgical treatment is preceded by hospital admission. With the ‘initial hospital admission’, we meant the in-hospital postoperative period. Therefore, we changed the following sentence in the introduction section to: “In order to prevent long-term problems in the postoperative period following hospital discharge, it is important to detect mastectomy skin flap and DIEP flap necrosis as early as possible during the in-hospital postoperative period.” See introduction: page 2, paragraph 3.
Furthermore, we changed the following sentence following sentence in the discussion section: “HSI has the potential to facilitate early and accurate debridement of necrotic tissue during the in-hospital postoperative period in order to prevent long-term wound problems in the postoperative period.” See Discussion: page 11, paragraph 3.
“Acceptor site” should be replaced with “recipient site”
We changed ‘Acceptor vessels’ to ‘recipient vessels’ and ‘acceptor site’ to ‘recipient site’ in the materials and methods section. See Materials and Methods: page 3, paragraph 3 and 4, respectively.
“…..and wore pressure stockings during the operation and for the time of (insufficient) immobilization.” Please clarify the meaning of this statement
All patients wore compression socks from the operation to adequate mobilisation (i.e., walking more than 30 minutes a day). The following sentence in the materials and methods section was changed to: “However, patients received prophylactic low-molecular-weight-heparin until discharge and wore compression socks during the operation up to the time of adequate mobilisation (i.e., walking more than 30 minutes a day).” See Materials and Methods: page 3, paragraph 4.
Reviewer 2 Report
The authors revised their manuscript and the title of the article according to the reviewer's suggestion. It is acceptable as a pilot study. I offer the authors for future study with larger number of patients, and please add the authors intention for the future study with large number of patients in conclusion.
Author Response
Answers to reviewer 2
Thanks again you for your comments. All your comments have been separately addressed, changes in the text are underlined. The changes are highlighted as ‘tracked changes’ in the manuscript.
The authors revised their manuscript and the title of the article according to the reviewer's suggestion. It is acceptable as a pilot study.
The title of the article is revised again according to the suggestion of reviewers 1. See title: “Perioperative hyperspectral imaging to assess mastectomy skin flap and DIEP flap perfusion in immediate autologous breast reconstruction: a pilot study”.
I offer the authors for future study with larger number of patients, and please add the authors intention for the future study with large number of patients in conclusion.
We indeed have the intention to report a future study with larger number of patients in order to add information on HSI parameter values for DIEP flap necrosis. We added the following sentence to the discussion: “Although we believe that HSI could also be useful in DIEP flap monitoring, this hypothesis should be confirmed in future studies including a larger sample size in order to obtain objective data regarding DIEP flap necrosis.”. See Discussion: page 9, paragraph 2.
Reviewer 3 Report
The authors have successfully addressed the comments raised by the reviewers.
Author Response
Thank you very much for your kind comment.
Round 3
Reviewer 1 Report
Dear authors,
The concerns raised in the previous round of review were correcly addressed.
There is a minor comment.
"diep flap necrosis was not observed" this data should be provided also in results of the manuscript, not only in abstract
Author Response
Answers to reviewer 1
Once again, thanks for your comment. Please see our reply below (changes in the text are underlined). The changes are highlighted as ‘tracked changes’ in the manuscript.
The concerns raised in the previous round of review were correctly addressed. There is a minor comment. "DIEP flap necrosis was not observed" this data should be provided also in results of the manuscript, not only in abstract.
We agree that this data should also be provided in the results section. In the previous version, we wrote “No necrosis was observed in ROI-2 or ROI-3”. For clarification, this sentence was removed. In addition, we added the following sentence to the results section: “In this study, DIEP flap (ROI-2) necrosis and necrosis of the medial side of the mastectomy skin flap (ROI-3) was not observed. However, three patients presented with mastectomy skin flap necrosis in ROI-1.” See Results, page: 5, paragraph 2.